# Genetic characterization of *Bacillus anthracis* strains circulating in Italy from 1972 to 2018

**Valeria Rondinone**[☯]**, Luigina Serrecchia**[☯]**, Antonio Parisi**[☯]**, Antonio Fasanella**[☯]**, Viviana Manzulli**[☯]**, Dora Cipolletta**[☯]**, Domenico Galante**[ID]**\***[☯]

Anthrax Reference Institute of Italy, Experimental Zooprophylactic Institute of Apulia and Basilicata Regions, Foggia, Italy

[☯] These authors contributed equally to this work.
\* domenico.galante@izspb.it

## Abstract

In Italy anthrax is an endemic disease, with a few outbreaks occurring almost every year. We surveyed 234 *B. anthracis* strains from animals (n = 196), humans (n = 3) and the environment (n = 35) isolated during Italian outbreaks in the years 1972–2018. Despite the considerable genetic homogeneity of *B. anthracis*, the strains were effectively differentiated using canonical single nucleotide polymorphisms (CanSNPs) assay and multiple-locus variable-number tandem repeat analysis (MLVA). The phylogenetic identity was determined through the characterization of 14 CanSNPs. In addition, a subsequent 31-loci MLVA assay was also used to further discriminate *B. anthracis* genotypes into subgroups. The analysis of 14 CanSNPs allowed for the identification of four main lineages: A.Br.011/009, A.Br.008/011 (respectively belonging to A.Br.008/009 sublineage, also known Trans-Eurasian or TEA group), A.Br.005/006 and B.Br.CNEVA. A.Br.011/009, the most common subgroup of lineage A, is the major genotype of *B. anthracis* in Italy. The MLVA analysis revealed the presence of 55 different genotypes in Italy. Most of the genotypes are genetically very similar, supporting the hypothesis that all strains evolved from a local common ancestral strain, except for two genotypes representing the branch A.Br.005/006 and B.Br.CNEVA. The genotyping analysis applied in this study remains a very valuable tool for studying the diversity, evolution, and molecular epidemiology of *B. anthracis*.

## Introduction

Anthrax is a non-contagious zoonotic disease affecting a broad range of animal species including humans. *Bacillus anthracis*, the etiological agent of anthrax, forms highly resistant spores that can persist in the environment for several decades [1]. Domestic and wild ruminants are species most susceptible to anthrax [2]. Animals are infected during grazing in areas contaminated with anthrax spores, while humans can contract the disease by contact with anthrax-infected animals or anthrax-contaminated animal products. Most frequently this involves employment in specific high risk occupation; such a farmer, butcher, tanner, wool carder, shearer and veterinarian. Exposure most commonly occurs during the skinning and butchering of cattle that are sick or dead because of anthrax [3]. Three forms of anthrax occur

**Data Availability Statement:** All relevant data are within the paper and its Supporting Information file.

**Funding:** The study was funded by Italian Ministry of Health, by a research program "Ricerca corrente IZS PB 005/15 RC" (www.salute.gov.it) to AF. The funder had no role in study design, data collection and analysis, decision to publish, or preparation of the manuscript.

**Competing interests:** The authors have declared that no competing interests exist.

in humans, depending on exposure type: cutaneous (which is usually non-fatal), gastrointestinal, and inhalational, both of which can be potentially fatal [4]. Recently, a fourth form of the disease was reported in drug users who inject drugs contaminated with anthrax spores [5]. Further, since it is relatively easy and inexpensive to obtain *B. anthracis*, the bacterium is one of the preferred pathogenic agents for use as a bacteriological weapon in bio-terrorist attacks [6]. In Italy, anthrax is typically a sporadic disease, particularly occurring during the summer (with a few exceptions) in the central and southern regions, and in the major islands, where it almost exclusively affects animals at pasture [7]. Between 1972 and 2018, approximately 200 outbreaks of animal anthrax were recorded (unpublished data). Very rarely, anthrax infection takes the form of an epidemic-like disease, characterized by outbreaks in limited areas involving a great number of animals. In Italy, two major epidemic-like anthrax outbreaks have been reported: one during the summer of 2004 in Basilicata, and one during the summer of 2011, in an area between Basilicata and Campania [8, 9]. Molecular tools, such as the canonical SNPs assay (CanSNPs) and multiple-locus variable-number tandem repeat analysis (MLVA) are highly effective for differentiating *B. anthracis* strains. The overall goal of this study was to utilize SNP analysis to establish the phylogenetic relationship between the *B. anthracis* strains examined, and further discriminate them in the context of the MLVA assay, in order to examine correlations among the *B. anthracis* isolates associated with the Italian anthrax outbreaks and to assess genetic diversity at regional and broader scales.

## Materials and methods

### Ethics statement

The animal and environmental strains used in the current study were isolated at the Anthrax Reference Institute of Italy (Ce.R.N.A.), a public laboratory mandated by the Italian Ministry of Health to confirm diagnosis of all animal anthrax cases in Italy. During outbreaks, samples were collected by the veterinary services of the Ministry of Health. Specific permission for soil sampling was not required. *B. anthracis* DNAs from anthrax human cutaneous cases were also included in the current study; two came from the "San Carlo" Hospital, Department of Infectious Disease, Potenza, Italy, and one from the "L. Spallanzani" National Institute for Infectious Disease, Rome, Italy [10].

### Bacterial strains

A collection of 234 *B. anthracis* strains, including 196 strains isolated from animal and 35 from the environment, isolated during Italian anthrax outbreaks in the years 1972–2018, were analyzed in the current study (Table 1). Furthermore, 3 *B. anthracis* DNAs from anthrax human cutaneous cases were also analyzed.

**Table 1. Overview of *Bacillus anthracis* isolates from the years 1972–2018 analyzed in the current study.**

| Sample type | Source | No. of isolates | Regions |
|---|---|---|---|
| Environmental samples | Water | 3 | Tuscany |
| | Soil | 32 | Basilicata, Tuscany |
| Animal samples | Bovine | 101 | Basilicata, Campania, Lazio, Apulia, Sardinia, Sicily, Tuscany, Umbria, Veneto, Lombardy |
| | Caprine | 20 | Abruzzo, Basilicata, Calabria, Campania, Apulia, Sardinia, Trentino |
| | Deer | 7 | Basilicata |
| | Equine | 12 | Basilicata, Campania, Apulia |
| | Ovine | 53 | Basilicata, Campania, Lazio, Apulia, Sicily |
| | Swine | 3 | Basilicata |
| Human samples (DNAs) | Human | 3 | Basilicata, Lazio |

## DNA extraction

*B. anthracis* strains were seeded on 5% sheep blood agar plates and then incubated at +37˚C for 24 h. Bacterial DNA was extracted using the DNAeasy Blood and Tissue kit (Qiagen, Hilden, Germany), following the protocol for Gram-positive bacteria. All manipulations of *B. anthracis* strains were performed in a biosafety level 3 laboratory at the Experimental Zooprophylactic Institute of Apulia and Basilicata Regions in a class II type A 2 biosafety cabinet.

## Real-time polymerase chain reaction (PCR) assay

Molecular identification of *B. anthracis* was performed using qualitative real-time PCR. The method is based on the amplification of specific DNA sequences using three pairs of specific primers [11] as follows: R1/R2 primers, specific for the BA813 gene located on the *B. anthracis* chromosome; PAG 23/24 primers, specific for the protective antigen (PA) gene located on the virulence plasmid pXO1; and CAP 57/58 primers, specific for the capsule (CAP) gene located on the virulence plasmid pXO2. Each 20 µl reaction mixture contained 1x Sso Advanced TM SYBR® Green Supermix (BIORAD), 300 nM each forward and reverse primer, and approximately 10 ng DNA template. The amplification was performed using the CFX Connect Real Time PCR Detection System (BIORAD). A melting curve was generated at 0.5˚C increments between 65˚C and 95˚C, and was analyzed by CFX Manager TM Software, Version 3.0 (BIORAD).

## CanSNP analysis

CanSNP profiles were obtained using 13 allelic discrimination assays involving specific oligonucleotides and probes, as described by Van Ert et al. [12]. Each 10 µl reaction mixture contained 1x TaqMan Genotyping Master Mix (Applied Biosystems, Foster City, CA, USA), 250 nM probe, 600 nM each of forward and reverse primer, and approximately 10 ng DNA template. For all assays, the thermal cycling parameters used were as follows: 10 min at 95˚C, followed by 40 cycles of 15 s at 95˚C and 1 min at 60˚C. Endpoint fluorescent data were acquired by using the ABI 7900HT instrument. The CanSNPs data were compared with the data for 12 recognized sublineage or subgroups. The 14th SNP was detected using a High Resolution Melting (HRM) assay for a specific A.Br.011 CanSNP [13,14]. Position 2,552,486, based on the Ames Ancestor genome (NC_007530.2), was analyzed. Amplification was performed using the CFX Connect Real Time PCR Detection System (BIORAD) and Precision Melt Supermix (BIORAD). The reaction mixture contained 0.2 µM of each primer and 1x Precision Melt Supermix (BIORAD) in a 20 µl final volume. The following cycling parameters were used: 2 min at 95˚C, were followed by 35 cycles of 10 s at 95˚C and 30 s at 60˚C. The samples were then heated to 95˚C for 30 s, cooled down to 60˚C over 1 min, and then heated from 65˚C to 95˚C at a rate of 0.5˚C/s. High-resolution melting data were analyzed using Precision Melt Analysis Software (BIORAD).

## 31-loci MLVA analysis

For the 31-marker MLVA, 5' fluorescently labeled oligonucleotides (6-FAM, VIC, NED and PET), specifically selected for variable number tandem repeats (VNTR) analysis were used. Twenty-seven chromosomal VNTR loci (vrrA, vrrB1, cg3, vrrB2, vntr19, vrrC1, vrrC2, vntr32, vntr12, vntr35, vntr23, bams03, bams05, bams13, bams15, bams21, bams22, bams23, bams24, bams25, bams28, bams30, bams31, bams34, bams44, bams51 and bams53) and four plasmid loci (vntr16, vntr17, pxO1 and pxO2) [12, 15–18] were analyzed. The MLVA assay involved preparation of two singleplex and nine multiplex reactions, in a final volume of 15 µl. Each reaction mixture contained the following: 1x PCR reaction buffer (Qiagen, Hilden, Germany); 3 mM MgCl$_2$, 0.2 mM for each dNTP; 1 U Hot Star Plus Taq DNA polymerase (Qiagen,

Hilden, Germany), the appropriate concentration of each primer (as described in Table 2); and approximately 10 ng DNA template.

The following PCR cycling program was used for the two singleplex reactions and for multiplex reactions 1 and 2: 5 min at 95˚C; followed by 35 cycles of 30 s at 94˚C, 30 s at 60˚C, and 30 s at 72˚C, with a final step of 5 min at 72˚C. The following amplification program was used for multiplex reactions 3: 5 min at 95˚C, followed by 35 cycles of 30 s at 94˚C, 30 s at 54˚C, 45 s at 72˚C, and 5 min at 72˚C. The following amplification program was used for multiplex reaction 4: 5 min at 95˚C, followed by 35 cycles of 30 s at 94˚C, 45 s at 56˚C, 1 min at 72˚C, and 5 min at 72˚. The following amplification program was used for multiplex reaction 5: 5 min at 95˚C, followed by 35 cycles of 30 s at 94˚C, 45 s at 59˚, 1 min at 72˚C, and 5 min at 72˚C. The following amplification program was used for multiplex reactions 6 to 9: 5 min at 94˚C, followed by 35 cycles of 1 min at 94˚C, 90 s at 60˚, 90 s at 72˚C, and 15 min at 72˚C.

## Automated genotype analysis

The MLVA PCR products were diluted 1:80 and analyzed by capillary electrophoresis using the ABI Prism 3130 genetic analyzer (Applied Biosystems) and 0.25 μl GeneScan 1200, and were sized by using Gene Mapper 4.0 (Applied Biosystems Inc.). Assignment of the sizes and corresponding repeating unit numbers for each locus was performed using the following strains as references: Ames Ancestor (NCBI Reference Sequence: NC_007530.2, chromosome), pXO1 (NCBI Reference Sequence: NC_007322.2, plasmid), and pXO2 (NCBI Reference Sequence: NC_007323.2, plasmid). Data were analyzed using conventional values proposed in the updated version of the 2016 *Bacillus anthracis* MLVA database, available at MLVAbank (http://mlva.u-psud.fr/). A phylogram was derived by clustering with the unweighted pair group method with arithmetic means (UPGMA), using 'categorical' character table values. All markers were given equal weight, irrespective of the number of repeats.

The discriminatory ability of the MLVA technique was determined by calculating the discriminatory index (D) for 234 typed strains. The discriminatory power of each VNTR was estimated by the number of alleles detected and the allele diversity using BioNumerics 7.6 software (Applied Maths, Belgium) [19].

## Results

### Real Time PCR, CanSNPs and MLVA analysis of anthrax strains

All the analyzed strains tested positive after the PCR amplification of chromosomal, plasmid pXO1 (toxins coding) and pXO2 (capsule formation) targets. The analysis of 13 classical

**Table 2. Primer concentration for each set of marker in PCR reactions of MLVA analysis.**

| PCR Reactions | Primers conc. [μM] |
|---|---|
| Singleplex 1 | vrrC1 [0.2 μM] |
| Singleplex 2 | vrrC2 [0.2 μM] |
| Multiplex 1 | vrrA, vrrB1 [0.2 μM]; CG3 [0.4 μM] |
| Multiplex 2 | vrrB2 [0.25 μM]; pXO1 [0.3 μM]; pXO2 [0.1 μM] |
| Multiplex 3 | vntr12 [0.25 μM]; vntr19 [0.2 μμM]; vntr35 [0.2 μM] |
| Multiplex 4 | vntr16 [0.25 μM]; vntr23 [0.2 μM] |
| Multiplex 5 | vntr17 [0.1 μM]; vntr32 [0.4 μM] |
| Multiplex 6 | bams03 [0.8 μM]; bams05 [0.2 μM]; bams15, bams44 [0.5 μM] |
| Multiplex 7 | bams21 [0.4 μM]; bams24, bams25 [0.3 μM]; bams34 [0.2 μM] |
| Multiplex 8 | bams13 [0.3 μM]; bams28 [0.15 μM]; bams31, bams53 [0.6 μM] |
| Multiplex 9 | bams22, bams51 [0.3 μM]; bams23 [0.2 μM]; bams30 [0.6 μM] |

CanSNPs revealed that 231 analyzed strains belonged to the sublineage A.Br. 008/009, also known as Trans-Eurasian (TEA) group. The TEA group was established in southern and eastern Europe and represents the dominant subgroup in Italy, Bulgaria, Hungary and Albania [7, 12, 20–22]. The analysis of an additional 14th CanSNP (A.Br.011), recently allowed for the differentiation of the A.Br.008/009 group into 2 subgroups. Accordingly, the analysis of the 14th CanSNP in the current study revealed that 207 of the 231 strains belonged to the main sublineage A.Br.011/009, while 24 strains belonged to the sublineage A.Br.008/011. All but one strain belonging to the latter sublineage were isolated in Sicily; one strain was isolated in Umbria. Further, one strain isolated in Veneto belonged to the main lineage A, sublineage A. Br.005/006, while two other strains, one from Veneto and one from Trentino, belonged to the main lineage B, sublineage B.Br.CNEVA.

MLVA based on the analysis of 31 VNTRs revealed 55 different genotypes, as shown in S1 Table, distributed in the Italian regions named GT-1 to GT-55, accordingly (Fig 1). The GT-14 genotype was the most common and was represented by 34 strains, mostly from Basilicata, Apulia, and Calabria. The GT-31 genotype was represented by 19 isolates: 16 from Tuscany, two from Apulia and one from Sardinia. The GT-26 and GT-27 genotypes were only isolated in the Basilicata and Campania regions. Other genotypes were characteristic for single regions, as showed in Table 3.

The number of different alleles ranged from 1 for bams21 and bams25 to 10 for bams15. Highest allelic diversities measured by Shannon Diversity Index (0.40632) was observed for the locus bams15 (Table 4). The relationship among the strains based on MLVA results is represented in Fig 2.

## Discussion

*Bacillus anthracis* is clonal in nature and often exhibits a high degree of genetic homogeneity due to the fact that is has a single stranded chromosome and reproduces asexually. This characteristic has traditionally made the discrimination of isolates for epidemiological purposes difficult. Furthermore the high survivability of spores in the soils, allowed *B. anthracis* to reproduce a relatively limited number of times during its evolution [23]. The 31-loci MLVA analysis carried out on 234 *B. anthracis* strains, isolated in Italy during the years 1972–2018, revealed the circulation of 55 *B. anthracis* genotypes. The performed CanSNPs analysis placed 53 of the 55 identified genotypes in a common cluster (TEA). The analysis of the classical 13 CanSNPs revealed that most of the analyzed strains (98%) belonged to the sublineage A. Br.008/009 (the TEA group), which is the most common group in Europe and Asia [15]. However, except for the genotypes of strains isolated in Umbria and some others isolated in Sicily belonging to sublineage A.Br.008/011, all strains belonged to the sublineage A.Br.011/009. Interestingly, genotype GT-54 isolated in Veneto was very different from the other characteristic Italian strains. CanSNPs analysis confirmed this observation placing this genotype in the branch A.Br.005/006. This branch is generally present in the central-southern Africa, although it was also identified in Europe [12, 24].Furthermore, genotype GT-55; B.Br.CNEVA, isolated in Veneto and Trentino is highly differentiated from most other Italian strains examined here. This genotype is widespread in Europe and found in France, Switzerland and Germany [12, 25, 26]. In Italy, the population of *B. anthracis* is mainly divided into two sublineages: A. Br.011/009, definitely the most common and A.Br.008/011 present only in Umbria and Sicily. Both these sublineages belong to the large TEA group (Fig 2). The TEA group A.Br.008/009 contains a *B. anthracis* subpopulation that is well adapted to the northern hemisphere and predominant in Europe, Russia, Kazakhstan, Caucasus and western China [12, 27]. It has also been detected in Africa [18, 28]. This group is thought to have given rise to the western north

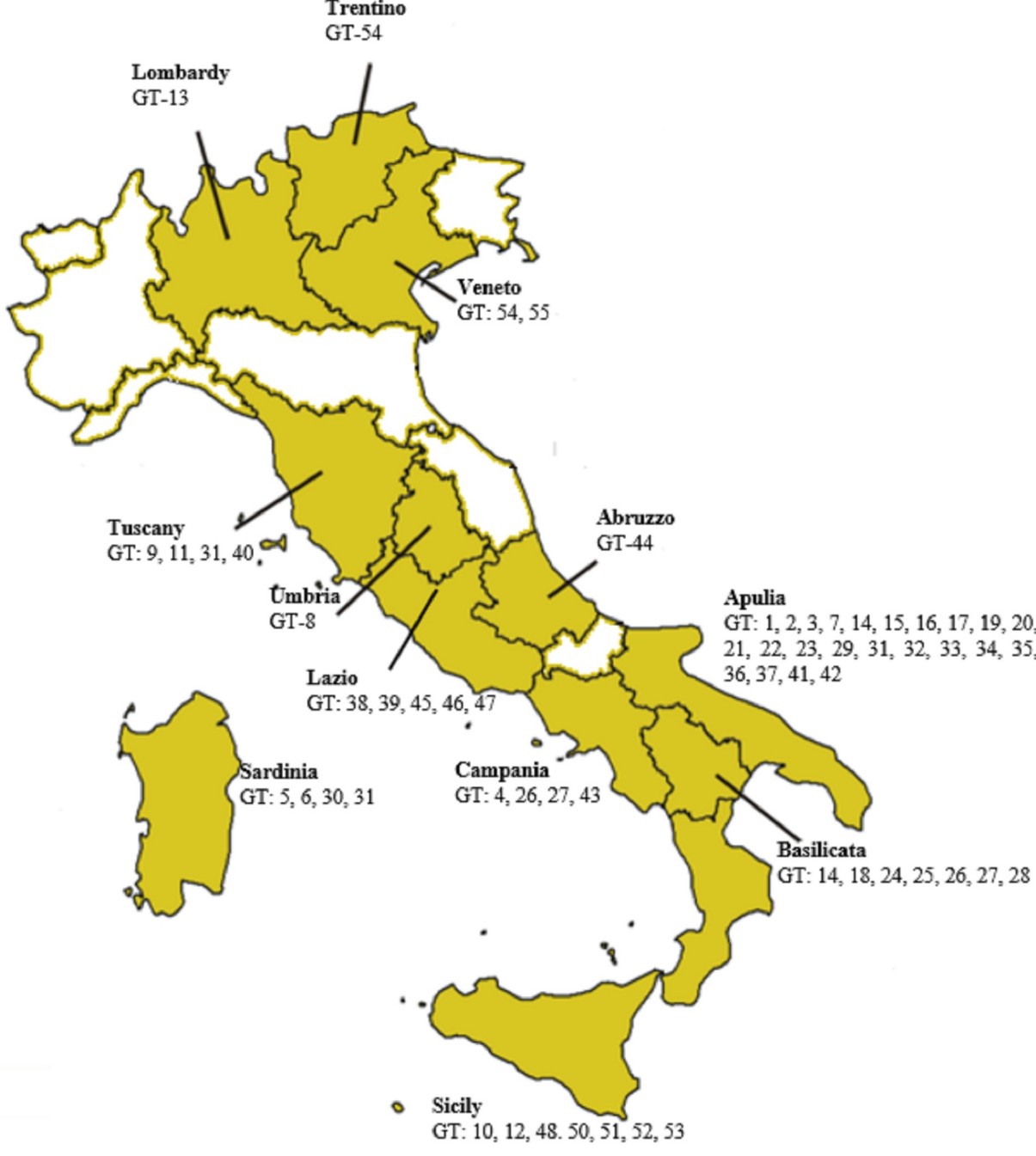

**Fig 1. The geographical distribution of 55 *Bacillus anthracis* genotypes in Italy.** Image modified from the "Map of Italy"; "World of Maps" Public Domain (*https://www.worldofmaps.net/europa/landkarten-und-stadtplaene-von-italien/landkarte-italien-administrative-bezirke-regioni.htm*).

American sublineage (A.Br.WNA), which is dominant in central Canada and much of the western USA. The presence of strains belonging to sublineages A.Br.008/011 and A.Br.011/009 might represent an effect of evolution on a common ancestral strain at the territorial level. In particular, A.Br.008/011 represents a rare and deep branching sublineage, also observed in Bulgaria, France and Turkey [29]. The spread of the TEA group to Europe and Asia is postulated to be linked to animal handling along the ancient East-West commercial routes of the

**Table 3. Distribution of *Bacillus anthracis* CanSNPs and genotypes isolated in Italy in the years 1972–2018.**

| Number of isolates | Regions | CanSNPs sublineage | Genotype |
|---|---|---|---|
| 1 | Apulia | A.Br. 011/009 | MLVA31-1 |
| 1 | Apulia | A.Br. 011/009 | MLVA31-2 |
| 1 | Apulia | A.Br. 011/009 | MLVA31-3 |
| 3 | Campania | A.Br. 011/009 | MLVA31-4 |
| 1 | Sardinia | A.Br. 011/009 | MLVA31-5 |
| 3 | Sardinia | A.Br. 011/009 | MLVA31-6 |
| 2 | Apulia | A.Br. 011/009 | MLVA31-7 |
| 1 | Umbria | A.Br. 008/011 | MLVA31-8 |
| 14 | Tuscany | A.Br. 011/009 | MLVA31-9 |
| 3 | Sicily | A.Br. 011/009 | MLVA31-10 |
| 1 | Tuscany | A.Br. 011/009 | MLVA31-11 |
| 3 | Sicily | A.Br. 011/009 | MLVA31-12 |
| 1 | Lombardy | A.Br. 011/009 | MLVA31-13 |
| 34 | Basilicata/Apulia/Calabria | A.Br. 011/009 | MLVA31-14 |
| 1 | Apulia | A.Br. 011/009 | MLVA31-15 |
| 2 | Apulia | A.Br. 011/009 | MLVA31-16 |
| 1 | Apulia | A.Br. 011/009 | MLVA31-17 |
| 1 | Basilicata | A.Br. 011/009 | MLVA31-18 |
| 1 | Apulia | A.Br. 011/009 | MLVA31-19 |
| 1 | Apulia | A.Br. 011/009 | MLVA31-20 |
| 1 | Apulia | A.Br. 011/009 | MLVA31-21 |
| 1 | Apulia | A.Br. 011/009 | MLVA31-22 |
| 1 | Apulia | A.Br. 011/009 | MLVA31-23 |
| 57 | Basilicata | A.Br. 011/009 | MLVA31-24 |
| 3 | Basilicata | A.Br. 011/009 | MLVA31-25 |
| 3 | Campania/Basilicata | A.Br. 011/009 | MLVA31-26 |
| 9 | Campania/Basilicata | A.Br. 011/009 | MLVA31-27 |
| 5 | Basilicata | A.Br. 011/009 | MLVA31-28 |
| 1 | Apulia | A.Br. 011/009 | MLVA31-29 |
| 1 | Sardinia | A.Br. 011/009 | MLVA31-30 |
| 19 | Tuscany/Apulia/Sardinia | A.Br. 011/009 | MLVA31-31 |
| 1 | Apulia | A.Br. 011/009 | MLVA31-32 |
| 1 | Apulia | A.Br. 011/009 | MLVA31-33 |
| 5 | Apulia | A.Br. 011/009 | MLVA31-34 |
| 6 | Apulia | A.Br. 011/009 | MLVA31-35 |
| 2 | Apulia | A.Br. 011/009 | MLVA31-36 |
| 1 | Apulia | A.Br. 011/009 | MLVA31-37 |
| 1 | Lazio | A.Br. 011/009 | MLVA31-38 |
| 1 | Lazio | A.Br. 011/009 | MLVA31-39 |
| 1 | Tuscany | A.Br. 011/009 | MLVA31-40 |
| 1 | Apulia | A.Br. 011/009 | MLVA31-41 |
| 1 | Apulia | A.Br. 011/009 | MLVA31-42 |
| 1 | Campania | A.Br. 011/009 | MLVA31-43 |
| 1 | Abruzzo | A.Br. 011/009 | MLVA31-44 |
| 2 | Lazio | A.Br. 011/009 | MLVA31-45 |
| 1 | Lazio | A.Br. 011/009 | MLVA31-46 |
| 5 | Lazio | A.Br. 011/009 | MLVA31-47 |

(*Continued*)

**Table 3.** (Continued)

| Number of isolates | Regions | CanSNPs sublineage | Genotype |
|---|---|---|---|
| 3 | Sicily | A.Br. 008/011 | MLVA31-48 |
| 1 | Sicily | A.Br. 008/011 | MLVA31-49 |
| 2 | Sicily | A.Br. 008/011 | MLVA31-50 |
| 9 | Sicily | A.Br. 008/011 | MLVA31-51 |
| 7 | Sicily | A.Br. 008/011 | MLVA31-52 |
| 1 | Sicily | A.Br. 008/011 | MLVA31-53 |
| 1 | Veneto | A.Br. 005/006 | MLVA31-54 |
| 2 | Trentino/Veneto | B.Br. CNEVA | MLVA31-55 |

Silk Road [30]. In the current study, strains belonging to the B.Br.CNEVA lineage were iso-lated in a relatively small area of north-eastern Italy. The relatively low diversity between the two strains demonstrated in the current study is consistent with a single introduction event of

**Table 4. Shannon Diversity Index and allele numbers of MLVA markers with respect to the collection investigated.**

| Locus | No. alleles | Diversity Index (Shannon) |
|---|---|---|
| vrrA | 4 | 0.172297 |
| vrrB1 | 2 | 0.021373 |
| vrrB2 | 3 | 0.073064 |
| vrrC1 | 2 | 0.021373 |
| vrrC2 | 2 | 0.082347 |
| CG3 | 2 | 0.02979 |
| pXO1aat | 4 | 0.344872 |
| pXO2at | 4 | 0.118086 |
| vntr32 | 3 | 0.033334 |
| bams03 | 2 | 0.021373 |
| bams05 | 5 | 0.08735 |
| bams13 | 5 | 0.1482 |
| bams15 | 10 | 0.40632 |
| bams21 | 1 | 0 |
| bams22 | 3 | 0.09788 |
| bams23 | 4 | 0.06145 |
| bams24 | 4 | 0.208345 |
| bams25 | 1 | 0 |
| bams28 | 2 | 0.23682 |
| bams30 | 6 | 0.11232 |
| bams31 | 7 | 0.224167 |
| bams34 | 3 | 0.030103 |
| bams44 | 2 | 0.147596 |
| bams51 | 5 | 0.183046 |
| bams53 | 3 | 0.021602 |
| vntr12 | 4 | 0.08852 |
| vntr16 | 5 | 0.219688 |
| vntr17 | 4 | 0.215683 |
| vntr19 | 2 | 0.234608 |
| vntr23 | 2 | 0.0708 |
| vntr35 | 2 | 0.159057 |

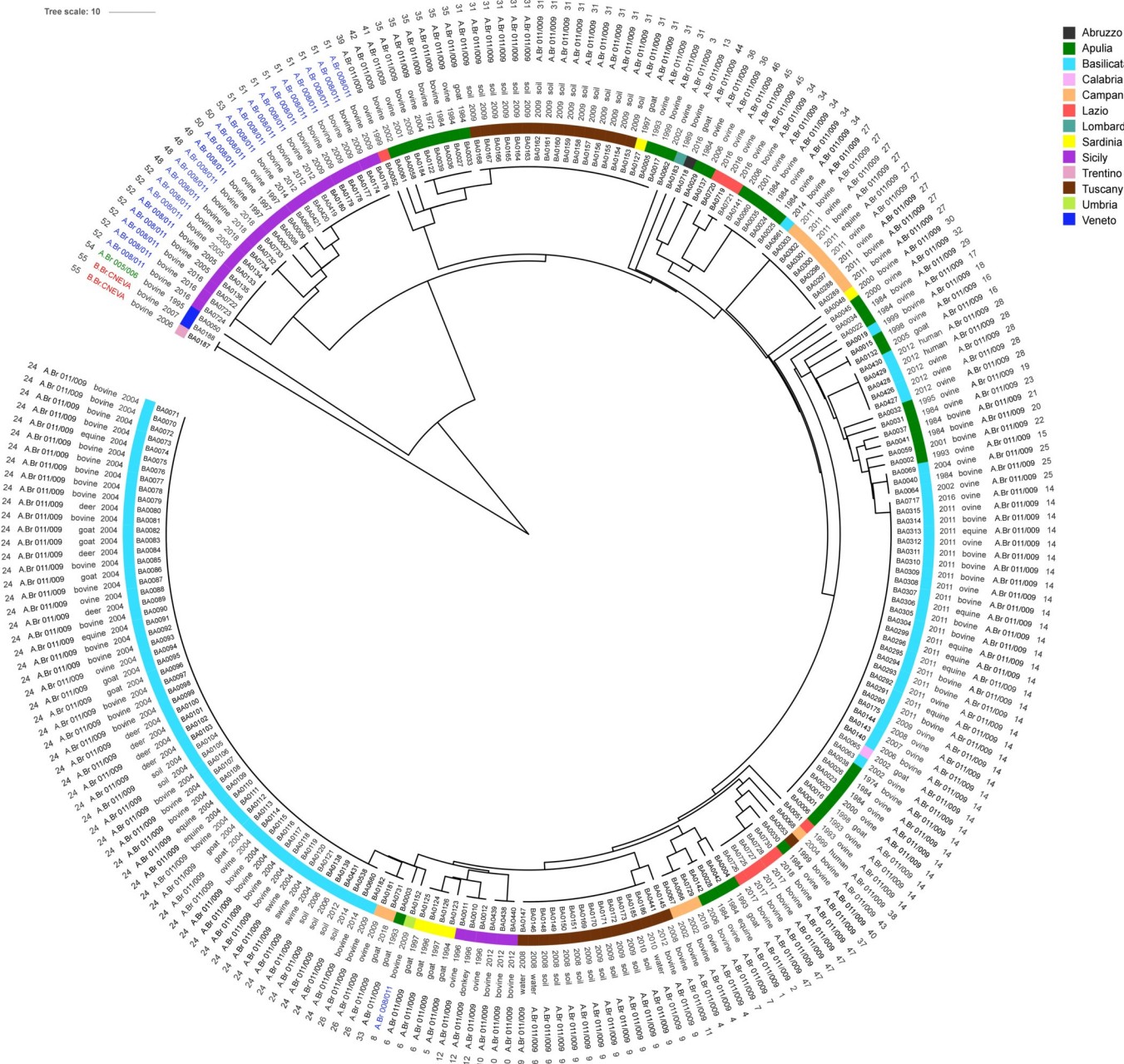

**Fig 2. A UPGMA phylogram of MLVA profiles.** The phylogram was built using BioNumerics 7.6 software (Applied Maths, Belgium). The visualization and the annotation of the genetic distances were performed using the web-based tool Interactive Tree of Life (iTOL). Circling the phylogram from the external to internal region are: genotype number, sublineage, species, year, regions (differently colored) of isolation and identification number of each analyzed strain.

Legend (regions):
- Abruzzo
- Apulia
- Basilicata
- Calabria
- Campania
- Lazio
- Lombardia
- Sardinia
- Sicily
- Trentino
- Tuscany
- Umbria
- Veneto

the B.Br.CNEVA lineage into the country, followed by ecological establishment and progressive *in situ* differentiation around the Italian Alps area [21]. Consistent with this hypothesis, the Italian strains form a cluster that is distinct from the other European B.Br.CNEVA strains. Identification of one A.Br.005/006 strain in Italy could be associated with the trade exchanges dating back when city states competed for trade and commerce throughout the Mediterranean [7]. This subgroup is well represented in Africa, but rare in Europe [12]. It is therefore quite surprising that past importations of ill or dying animals or spore-infected items from Africa,

the Middle East, or even Asia, did not have a greater impact on the genetic structure of *B. anthracis* in the region. The higher variety of *B. anthracis* genotypes identified in southern Italy relative to genotypes from other Italian regions may be explained by the differences in the breeding systems between northern and southern Italy. In southern Italy, many livestock farmers use extensive farming methods, which increases the chances of grazer exposure to historical spore sites and deposits. The possibility of exposure is lower in northern Italy because most livestock farmers use intensive breeding systems. Another observation from the current study was that the neighboring regions share just a few genotypes. In particular, the GT-24 genotype was present in Apulia, Basilicata and Calabria; the GT-26 and GT-27 genotypes were identified in Basilicata and Campania; and the GT-55 genotype was identified in Veneto and Trentino. Noteworthy and difficult to explain is the dislocation of genotype GT-31, identified in Apulia, Tuscany and Sardinia. These are not neighboring regions; on the contrary, they are quite far from one another. Also in this national scenario one of the explanations could be the trade of animals or animal products within the country over the years. Nevertheless, since most genotypes are exclusive to each region, it appears that Italian *B. anthracis* strains may be autochthonous for a single territory. Interestingly, genotypes exclusive to specific regions were detected especially in Sicily and Sardinia, probably because of low animal movements between these islands and the rest of Italy. The analysis of chromosomal and plasmid hypervariable regions using such methods as MLVA constitutes a valuable approach for studying the diversity, evolution and molecular epidemiology of *B. anthracis*. Therefore, MLVA is a valid method that enables the understanding of the distribution of *B. anthracis* within a country.

## Supporting information

**S1 Table. Allele distribution of the 55 genotypes identified using 31 VNTR analysis.** (XLSX)

## Acknowledgments

We thank Angela Aceti, Michela Iatarola, Elena Poppa and Francesco Tolve for the technical support.

## Author Contributions

**Conceptualization:** Valeria Rondinone, Luigina Serrecchia, Antonio Fasanella, Domenico Galante.

**Data curation:** Luigina Serrecchia, Antonio Parisi, Viviana Manzulli, Domenico Galante.

**Formal analysis:** Valeria Rondinone, Luigina Serrecchia, Viviana Manzulli.

**Funding acquisition:** Antonio Fasanella.

**Investigation:** Valeria Rondinone, Luigina Serrecchia, Antonio Fasanella, Viviana Manzulli, Dora Cipolletta.

**Methodology:** Valeria Rondinone, Luigina Serrecchia, Viviana Manzulli, Dora Cipolletta, Domenico Galante.

**Resources:** Antonio Fasanella.

**Software:** Antonio Parisi.

**Supervision:** Antonio Parisi, Antonio Fasanella, Domenico Galante.

**Visualization:** Domenico Galante.

**Writing – original draft:** Valeria Rondinone, Luigina Serrecchia, Domenico Galante.

**Writing – review & editing:** Valeria Rondinone, Antonio Parisi, Viviana Manzulli, Domenico Galante.

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
