## [Decision Letter · Decision Letter 0]

21 Sep 2019

PONE-D-19-23489

Genetic characterization of Bacillus anthracis strains circulating in Italy from 1972 to 2018

PLOS ONE

Dear Dr. Galante,

Thank you for submitting your manuscript to PLOS ONE. After careful consideration, we feel that it has merit but does not fully meet PLOS ONE’s publication criteria as it currently stands. Therefore, we invite you to submit a revised version of the manuscript that addresses the points raised during the review process.

The reviewers provide important suggestions on clarifying both the methodology used and the interpretation of results regarding the ancestry of strains within Italy, and the occurrence/movements of these sub-groups regionally/globally.

We would appreciate receiving your revised manuscript by Nov 05 2019 11:59PM. To enhance the reproducibility of your results, we recommend that if applicable you deposit your laboratory protocols in protocols.io, where a protocol can be assigned its own identifier (DOI) such that it can be cited independently in the future. For instructions see: http://journals.plos.org/plosone/s/submission-guidelines#loc-laboratory-protocols

We look forward to receiving your revised manuscript.

Kind regards,

Wendy C. Turner

Academic Editor

PLOS ONE

Journal Requirements:

2. We ask that you please explain in more detail the human samples used in this study:

i) Please clarify whether these are bacterial isolates only or some type of tissue or human fluid/excretion sample.

ii) If they are human tissue/excretion samples, please explain how they were collected, who collected them, and whether they were anonymized when you received them.

iii) Please also explain whether you obtained Institutional Review Board approval or ethics committee approval for the use of human tissue/material.

In addition, we noticed you have some minor occurrence of overlapping text with previous publications:

https://scielosp.org/article/aiss/2014.v50n2/192-195/en/

http://dx.doi.org/10.4236/ojvm.2012.22012

which needs to be addressed. In your revision ensure you cite all your sources (including your own works), and quote or rephrase any duplicated text outside the methods section. Further consideration is dependent on these concerns being addressed.

Reviewers' comments:

Reviewer's Responses to Questions

**Comments to the Author**

1. Is the manuscript technically sound, and do the data support the conclusions?

Reviewer #1: Partly

Reviewer #2: Yes

2. Has the statistical analysis been performed appropriately and rigorously? 

Reviewer #1: N/A

Reviewer #2: N/A

3. Have the authors made all data underlying the findings in their manuscript fully available?

Reviewer #1: No

Reviewer #2: Yes

4. Is the manuscript presented in an intelligible fashion and written in standard English?

Reviewer #1: Yes

Reviewer #2: No

5. Review Comments to the Author

Reviewer #1: Thank you for an interesting paper. The description of the genotypic distribution of B. anthracis isolates across Italy adds to the body of work which highlights the diversity of this pathogen internationally.

That said, please be careful of your repetition of the word "results" in regards to canSNP lineages. I have made some editorial suggestions in comment boxes. Also, please check your references, especially with regards to the PCR's used. You referred to real time PCR and then referenced Fasanella et al. which describes conventional PCR. You must credit the designers of the primers used in your methodology references. You did not refer to Patra and Ramisse for the pag, lef, cap and BA813 primers. You also did not list which primers exactly were used. If indeed you did use qPCR in diagnostics this study, please state the chemistry employed; if not...adjust the method description accordingly.

Would it be possible to add more detail surrounding the history of the isolates? Your references were only in relation to the epizootics involving tabanids/horseflies. More context would help flesh out your discussion and provide more evidence for solid conclusions. As it stands, you cannot conclude a common ancestor of strains using a limited panel of canSNP's and MLVA. If there is whole genome data to support this, you should refer to it. Your speculation around the introduction of the A.Br.005/006 also needs stronger evidence before you can call it a viable hypothesis. Otherwise, just call it speculation.

Reviewer #2: The manuscript is sound, although they need to drop the term phylogenetic tree from their UPGMA analysis. Not much analysis in this paper overall, mostly descriptive in scope.

Really needs some serious grammatical work.

6. PLOS authors have the option to publish the peer review history of their article (what does this mean?). If published, this will include your full peer review and any attached files.

Reviewer #1: No

Reviewer #2: No

---

## [Author Response · Author response to Decision Letter 0]

7 Nov 2019

Dear Editorial Office,

we made and submitted the corrections required by the Journal and by the reviewers. The manuscript was revised by a professional English proofreading service as the revievers asked.

Thank you for the precious suggestions and observations.

We submitted the revised manuscript following PLOS ONE's style requirements and we replaced the overlapping text with previous publications as you asked. 

The human samples we used in this study are just 3 DNAs coming from people affected by cutaneous anthrax for genotyping analyses. We received them in anonymous form. We described the origin in the manuscript.

Reviewer 1: We answered the questions directly in the pdf file he sent and in a separate word file. 

However we changed the reference he asked and we better described the primers used in the manuscript. We disecribed the Real Time PCR we used with Sybr Green technology.

Most of the isolates we tested are all the isolates we collected all over the years during the Italian anthrax outbreaks in animals. The environmental samples were taken from soils and water following anthrax outbreaks. The references you are referring to are related to the description of the epidemic-like anthrax outbreaks occurred in Italy in Basilicata and Campania. They were not inserted for a description of all the samples we analyzed.

As suggested we reworded the sentence with this hypothesis and deleted part of It. 

Reviewer 2: The term "phylogenetic tree" was replaced by the term "phylogram" and diversity index analysis was later performed in the manuscript.

The manuscript was revised by a professional English proofreading service for grammar.

---

## [Decision Letter · Decision Letter 1]

4 Dec 2019

PONE-D-19-23489R1

Genetic characterization of Bacillus anthracis strains circulating in Italy from 1972 to 2018.

PLOS ONE

Dear Dr. Galante,

Thank you for submitting your manuscript to PLOS ONE. After careful consideration, we feel that it has merit but does not fully meet PLOS ONE’s publication criteria as it currently stands. Therefore, we invite you to submit a revised version of the manuscript that addresses the points raised during the review process.

The reviewers note edits and minor corrections still necessary, but otherwise, a much improved paper. Please address these final comments.

We would appreciate receiving your revised manuscript by Jan 18 2020 11:59PM. To enhance the reproducibility of your results, we recommend that if applicable you deposit your laboratory protocols in protocols.io, where a protocol can be assigned its own identifier (DOI) such that it can be cited independently in the future. For instructions see: http://journals.plos.org/plosone/s/submission-guidelines#loc-laboratory-protocols

We look forward to receiving your revised manuscript.

Kind regards,

Wendy C. Turner

Academic Editor

PLOS ONE

Reviewers' comments:

Reviewer's Responses to Questions

**Comments to the Author**

1. If the authors have adequately addressed your comments raised in a previous round of review and you feel that this manuscript is now acceptable for publication, you may indicate that here to bypass the “Comments to the Author” section, enter your conflict of interest statement in the “Confidential to Editor” section, and submit your "Accept" recommendation.

Reviewer #1: (No Response)

Reviewer #2: All comments have been addressed

2. Is the manuscript technically sound, and do the data support the conclusions?

Reviewer #1: Partly

Reviewer #2: Yes

3. Has the statistical analysis been performed appropriately and rigorously? 

Reviewer #1: Yes

Reviewer #2: Yes

4. Have the authors made all data underlying the findings in their manuscript fully available?

Reviewer #1: Yes

Reviewer #2: Yes

5. Is the manuscript presented in an intelligible fashion and written in standard English?

Reviewer #1: Yes

Reviewer #2: Yes

6. Review Comments to the Author

Reviewer #1: The document still requires a bit of editorial correction, although vastly improved from the previous round.

I still have issue with some of the conclusions made in the discussion portion of the paper. While it is true that the trade history of Italy has been extensive over the centuries, conclusions regarding the genotypes of this study (limited to 1972-2018) cannot be compared to the middle ages unless you have solid data to support it. By the logic of your hypothesis, all isolates can be linked to trade with southern Africa. The time scales with which such genotypic distribution could have occured cannot be determined using MLVA (which inherently includes homoplasy/homoplasticity) and a limited panel of SNPs. Only whole genome sequencing and comparative genomics which includes diverse lineages can provide enough support for such a narrative. There is enough data to recommend this paper without the inclusion of such wild conclusions on the origins of these isolates.

As to the inclusion of the Simpson's index: In my exprerience, the Shanon diversity better represents allelic diversity in VNTR loci. It will also make your work comparable to other papers.

It reads much better overall and the methods are also much improved.

Reviewer #2: Overall this MS has improved greatly from the previous version, although there is still extensive editing to be done in terms of grammar. I have tried to address all of these issues below but have likely still missed some. I would suggest the authors use the free program/web app Grammarly to specifically address the issue of the overuse of commas throughout. With the incorporation of the edits below I think this manuscript would be suitable for publication in PLOS One.

Line 51: remove s at the end of “occupations”

Line 53: add s at the end of “human”.

Line 55: change “a fourth disease form” to “a fourth form of the disease”

Line 57: change “for use as bacteriological” to “for use as a bacteriological”

Line 66: remove comma

Line 75: remove comma after “laboratory”

Line 130: Remove comma after “analysis” and replace “we used” with “were used”.

Line 139: Table 2: Make sure all rows are equally spaced (rows 6 through 9 are spaced farther apart then the rest)

Lines 157-158: The authors should state what the reference numbers refer to (ncbi or ena, and what specific database)

Line 159: Change “Phylogram”, to “A phylogram”.

Line 171: Why does sentence end mid-page?

Line 172: Remove comma

Line 173: Remove “or”

Line 174: Remove comma after Europe

Line 175: Remove both commas surrounding “14th”

Line 176: change “to” to “the”

Line 183: remove comma after “VNTRs”

Line 184: remove comma after “regions”

Line 198: Table 4: Change “Simpson’s Index of Diversity” to “Simpson's Diversity Index” and throughout.

Line 202: Remove extreme spacing between words

Line 207: Change “Around the phylogram are shown, from the external part to the internal part” to “Circling the phylogram from the external to internal region are”

Line 214: replace comma with “and”

Line 215: change “purpose” to “purposes”

Line 216-217: I’m not sure this sentence is true, and it is also grammatically incorrect. Does the high survivability of spores lead to genetic homogeneity? I don’t think so, I would argue that genetic homogeneity in this species is due to the fact that it has a single stranded chromosome and reproduces asexually. Persistence of spores in the environment complicate this but is not the defining feature that dictates homogeneity.

Line 217-219: This sentence should be reworded to reduce the number of commas and read more coherently.

Line 221: Remove comma

Line 225: Remove comma after “GT-54” and “Veneto”

Line 226: Remove comma after “observation”

Line 228: Remove comma after “GT-55”

Line 229: Remove comma after “Trentino”

Line 228-229: This sentence is confusing because your allocating a genotype to a genotype. I would reword this to something like: “Furthermore, genotype GT-55; B.Br.CNEVA, isolated in Veneto and Trentino is highly differentiated from most other Italian strains examined here.

Line 230: change “branch” to “genotype”, remove “mainly” and change “in particular” to “and found in”

Line 236: Change “this group gave rise” to “this group is thought to have given rise” and cite the paper that originally presented this hypothesis.

Line 238: Change “an effect of evolution of a common ancestral strain at territorial level” to “an effect of evolution on a common ancestral strain at the territorial level”

Line 240, change “as well as” to “and”

Line 241: change “seems” to “is postulated”

Line 247: add the word “strains” after B.Br.CNEVA

Line 248: remove the word “the”

Line 254: remove comma after “regions” and change “can” to “may”

Line 256: change “increasing the chances” to “which increases the chances”

Line 257: Change “this” to “the”

Line 263: Change “This” to “these”

Line 264: change space in front of period

7. PLOS authors have the option to publish the peer review history of their article (what does this mean?). If published, this will include your full peer review and any attached files.

Reviewer #1: No

Reviewer #2: Yes: Spencer A Bruce

---

## [Author Response · Author response to Decision Letter 1]

23 Dec 2019

Dear Editorial Office,

we made and submitted the corrections required by the second revision of the Journal and of the reviewers. 

The corrections are reported in the attached files "response to reviewers" and "revised manuscript with track changes".

Thank you for the further precious suggestions and observations.

---

## [Editor Report · Decision Letter 2]

2 Jan 2020

Genetic characterization of Bacillus anthracis strains circulating in Italy from 1972 to 2018.

PONE-D-19-23489R2

Dear Dr. Galante,

We are pleased to inform you that your manuscript has been judged scientifically suitable for publication and will be formally accepted for publication once it complies with all outstanding technical requirements.

With kind regards,

Wendy C. Turner

Academic Editor

PLOS ONE
---

## [Editor Report · Acceptance letter]

6 Jan 2020

PONE-D-19-23489R2 

Genetic characterization of *Bacillus anthracis* strains circulating in Italy from 1972 to 2018. 

Dear Dr. Galante:

I am pleased to inform you that your manuscript has been deemed suitable for publication in PLOS ONE. Congratulations! Your manuscript is now with our production department. 

With kind regards,

on behalf of

Dr. Wendy C. Turner 

Academic Editor

PLOS ONE